# Identification of Triploid Plants in Seed-Derived Progeny of Cultivated Olive

**DOI:** 10.3390/plants15010127

**Published:** 2026-01-01

**Authors:** Chenggong Lei, Guangmin Wu, Yingjia Liu, Chengdu Yang, Qianli Dai, Yingchun Zhu, Fa Xiao, Hengxing Zhu, Jiangbo Dang

**Affiliations:** 1Key Laboratory of Agricultural Biosafety and Green Production in the Upper Yangtze River, Ministry of Education, College of Horticulture and Landscape Architecture, Southwest University, 2 Tiansheng Road, Beibei District, Chongqing 400715, China; leichenggong808@163.com (C.L.); wuguangmin26@163.com (G.W.); zhuyingchun2000@163.com (Y.Z.); sylmx20@163.com (F.X.); 2Chongqing Key Laboratory of Forest Ecological Restoration and Utilization in the Three Gorges Reservoir Area, Chongqing Academy of Forestry, 106 Gaodianzi, Geleshan Subdistrict, Shapingba District, Chongqing 400036, China; daiqianli126@126.com; 3Guizhou Institute of Fruit Tree Science, 307 Southwest Ring Road, Huaxi District, Guiyang 550025, China; 13688595562@163.com; 4College of Resources and Environment, Southwest University, 2 Tiansheng Road, Beibei District, Chongqing 400715, China; ydc051118@126.com

**Keywords:** olive, triploid, flow cytometry, InDel marker

## Abstract

The large and hard olive pit adversely affects oil quality during traditional crushing, as seed- and pit-derived enzymes modify phenolic profiles and volatile compounds. Polyploid breeding offers a potential means to reduce pit size and improve processing traits, yet cultivated olive (*Olea europaea* L. subsp. *europaea*) is a strictly diploid species, and natural polyploids have not been previously documented. To evaluate the potential of triploids in olive improvement, we screened seed-derived progeny from multiple cultivars for polyploidy using flow cytometry and chromosome observation. One naturally occurring triploid seedling (‘Olive-3x’) was identified from a mixed lot of open-pollinated seeds. Whole-genome resequencing was used to develop 64 polymorphic InDel markers, and three markers indicated ‘Koroneiki’ as one putative parent of the triploid. Morphological and cytological analyses showed that the triploid exhibited typical polyploid characteristics, including thicker leaves and enlarged epidermal and palisade mesophyll cells compared with diploid controls. These findings provide the first evidence of a naturally occurring triploid in cultivated olive and show that triploids can arise within seed-derived progeny. The identified triploid plant and the developed markers offer useful resources for future studies on olive polyploidy and provide foundational resources for future research on olive polyploidy and cultivar improvement.

## 1. Introduction

*Olea europaea* L., commonly known as the olive, is a species in the *Oleaceae* family and the *Olea* genus. It is an important woody oilseed crop cultivated worldwide, with evidence of cultivation dating back to the Mediterranean region around 6000 years ago [1]. The olive oil extracted from its fruits is rich in various vitamins and unsaturated fatty acids, making it one of the most nutritionally beneficial oils for human health. It is widely used in the food industry and health-related fields [2,3,4]. Currently, olive trees are cultivated in more than 40 countries globally, with a distribution range that spans the Mediterranean coast, subtropical, temperate, and some cold regions [5].

When traditional crushing techniques are used for oil extraction, the olive pit and seed affect the phenolic compounds and volatile substances in olive oil through enzymatic activity, thereby altering its chemical and sensory properties and resulting in reduced oil quality and shortened shelf life. Consequently, there has been a strong drive toward the development of pit removal technologies and the upgrading of related equipment [6,7,8]. From a biological and breeding perspective, reducing pit and seed effects at the source represents an alternative and complementary approach.

Triploids, due to abnormalities in meiosis, often produce large fruits, small pits, or seedless fruits, which greatly benefit the reduction of pit and seed influence on olive oil quality, and improve the fruit utilization rate and processing convenience. Currently, triploids are widely used in fruit trees such as bananas [9], pears [10], loquats [11], and citrus [12]. In addition, some fruit trees with hard pits, such as jujube [13,14] and plums [15], also have triploid varieties. The triploid Japanese plum ‘Kiyo’ has large fruits and high quality [15]; the ‘Zanhuang Dazao’ jujube produces large fruits, and its variant ‘Zanshuo’, developed from a mutation of this cultivar, has an edible rate of up to 95.8% [16]. In addition, triploids possess stronger stress tolerance, including resistance to diseases, cold, and salinity [17]. Certain triploid apple and poplar genotypes outperform their diploid counterparts in either disease resistance or environmental stress tolerance, demonstrating the potential of triploid breeding to enhance the resilience of woody crops [18,19,20,21].

Cultivated olive (*O. europaea* subsp. *europaea*) and wild olive (*O. europaea* var. *sylvestris*) are both diploid species (2n = 2x = 46) [22]. To date, no natural polyploids have been reported in cultivated olives. However, natural polyploidy has been observed within the olive complex. For example, tetraploid and hexaploid individuals have been identified in subsp. *ceraciformis* and subsp. *maroccana*, respectively [23]. In addition, Besnard and Baali-Cherif reported the discovery of three triploid individuals in a natural population of subsp. *laperrinei* from the Hoggar region in Algeria, with a triploid frequency of 2.8% [24].

Gamma-ray irradiation has been shown to induce mixoploid mutants in cultivated olive, such as FC (from ‘Frantoio’) and LC (from ‘Leccino’) [25,26,27]. These mixoploid mutants served as starting material for further polyploid development. Triploid individuals (2n = 3x = 69) were obtained from seeds of unusually large fruits (almost twice the size of diploid fruits), and tetraploids (2n = 4x = 92) were isolated from FC through in vitro selection [28]. Subsequently, stable tetraploid mutants, FRM5-4n and LM3-4n, were derived from the shoot tips of FC and LC, respectively [29]. However, polyploid germplasm resources in olive remain extremely limited, which constrains comprehensive evaluation of their agronomic potential and practical application. Notably, whether naturally occurring triploids exist within seed-derived progenies of cultivated olive remains unclear, and efficient approaches for their identification and characterization have not yet been established.

Rugini et al. found that although the tetraploid mutant ‘Leccino’ (LM3-4n) exhibited lower fruit set rates and some variability in fruit size, its fruits were still significantly larger than those of the diploid mutant (LM3-2n) [29]. This finding suggests that artificially induced polyploids may exhibit certain limitations despite improvements in specific traits. In contrast, natural polyploids, having undergone homologous recombination and long-term natural selection, often display greater genetic stability, improved reproductive performance, enhanced environmental adaptability, and superior growth potential compared with artificially induced counterparts [30,31]. Furthermore, the 2n gamete pathway is widely recognized as an important mechanism underlying natural polyploid formation and has been successfully exploited in breeding programs of fruit crops such as citrus [32,33] and loquat [34]. However, corresponding studies and applications of natural polyploid screening based on seed-derived populations remain scarce in olive.

Molecular markers have been widely used in the analysis of olive germplasm evaluation, cultivar identification, and genetic relationship analysis [5,35,36,37]. Among them, Insertion-Deletion (InDel) markers, derived from length polymorphisms caused by nucleotide insertions or deletions, are characterized by wide genomic distribution, high density, stable inheritance, codominance, and good reproducibility [38,39]. In woody crops such as grape [40], apple [41], and citrus [42], InDel markers have proven effective for cultivar discrimination and parentage analysis. Despite these advantages, InDel markers have not yet been systematically developed or applied in olive, limiting their potential use in cultivar identification and genetic analysis. With the availability of the olive reference genome, the development and application of InDel markers provide new technical support for olive germplasm characterization and genetic improvement.

To address the lack of information on natural triploidy in seed-derived progenies of cultivated olive, this study employed flow cytometry combined with chromosome observation to screen for naturally occurring triploid plants. In parallel, InDel molecular markers were developed and applied to analyze genetic relationships and infer the maternal origin of the identified triploid. In addition, leaf morphological and anatomical traits associated with different ploidy levels were systematically characterized. Together, these results provide new insights into the discovery of natural polyploid olive germplasm and offer practical methodological references for ploidy identification and molecular marker-assisted studies in olive.

## 2. Results

### 2.1. Triploid Olive

Flow cytometry was used to analyze the ploidy level of the olive seedlings. Among the 2754 seedlings, a single triploid individual was identified from progeny derived from a mixed seed lot of 12 cultivars and was designated ‘Olive-3x’. Using ‘Koroneiki’ as the diploid control, mixed sample preparation was performed for analysis. Two distinct DNA fluorescence peaks were detected (Figure 1). The first peak was located at the horizontal coordinate of 6188.00, while the second peak was observed at 8861.00, corresponding to 1.43 times the position of the first peak. Chromosome counting further confirmed that this plant had 69 chromosomes, thereby verifying its triploid status (Figure 1).

### 2.2. Identification of the Potential Parents of the Triploid Olive Using InDel Molecular Markers

#### 2.2.1. Development of InDel Markers

Whole-genome resequencing was performed on 5 olive cultivars, yielding a total of 941,577,110 high-quality reads that were successfully mapped to the wild olive reference genome. On average, the proportion of bases with a quality score higher than Q20 was 96.18%, and that with a quality score higher than Q30 was 90.89%. The average GC content was 37.46%, the mean genome coverage was 92.70%, and the mean sequencing depth reached 18.03×.

A total of 76,400 InDels (≥15 bp) were identified, including 20,787 insertions and 55,613 deletions. These InDels were broadly distributed across all 23 chromosomes, with an average of 3322 InDels per chromosome. Chromosome 10 contained the highest number of InDels (6480), whereas chromosome 12 contained the fewest (1717). Most of the InDels were located in intergenic regions, while 1151 were detected within genes. Among the 727 exonic InDels, 454 resulted in frameshift mutations. In addition, 7767 InDels were located in the upstream or downstream regions of genes (Figure 2).

#### 2.2.2. Development and Validation of InDel Primers and Screening of Polymorphic Markers

Based on the genome resequencing data, a total of 8604 primer pairs were designed from InDel loci with insertion/deletion lengths ≥15 bp. From these, 460 loci evenly distributed across the 23 chromosomes were selected for primer design. Using genomic DNA of ‘Golden Leaves’ and ‘Arbequina I’ as templates, PCR amplification was performed. Electrophoresis results showed that 309 primer pairs produced clear bands. Subsequently, the polymorphism of these 309 primer pairs was validated in 15 olive cultivars (Table 1). PCR amplification and polyacrylamide gel electrophoresis revealed that 64 primer pairs (Appendix A) generated distinct polymorphic bands (Figure 3). Based on the polymorphism results, cluster analysis was conducted on the 15 cultivars (Appendix A), confirming the effectiveness of the developed markers.

Subsequently, the polymorphism of 309 InDel primer pairs was evaluated using ‘Olive-3x’ and 15 olive cultivars (Table 1). PCR amplification followed by polyacrylamide gel electrophoresis revealed that 64 primer pairs (Appendix A) generated clear and stable polymorphic bands (Figure 3). Cluster analysis based on these 64 polymorphic markers (Appendix A) effectively separated ‘Olive-3x’ and 15 olive cultivars, demonstrating that the developed InDel markers possess strong discriminatory power and are suitable for cultivar identification and parentage or diversity analysis.

#### 2.2.3. Analysis of the Potential Parentage of the Triploid

The triploid originated from a seedling population derived from mixed seeds of 12 cultivars (Table 1). To infer its potential parentage, homozygous loci of the triploid and those of the 15 cultivars were compared using 64 InDel markers. The results showed that a minimum combination of three InDel markers (OE03-5, OE05-12, and OE03-17) was sufficient to identify one potential parent of the triploid ‘Olive-3x’. As shown in Figure 4, amplification with marker OE03-5 indicated that no bands corresponding to ‘Olive-3x’ were detected in ‘Picual’, ‘Leccino’, ‘Coratina’, ‘Yuntai’, ‘Ezhi 8# I’ or ‘Ezhi 8# II’, excluding them as possible parents. Similarly, marker OE05-12 excluded ‘Picual’, ‘Leccino’, ‘Coratina’, ‘Frantoio’, ‘Yuntai’, ‘Pendolino’, ‘Golden Leaves’, ‘Ezhi 8# I’ and ‘Ezhi 8# II’ as possible parents. Marker OE03-17 excluded ‘Picual’, ‘Arbequina I’, ‘Arbequina II’, ‘Coratina’, ‘Yuntai’, ‘Arbosana I’, ‘Arbosana II’ and ‘Golden Leaves’ as possible parents. In contrast, all three primer pairs amplified the same bands in the genomic DNA of ‘Koroneiki’ as observed in ‘Olive-3x’. Therefore, ‘Koroneiki’ was identified as the most likely maternal parent of the triploid.

### 2.3. Morphological Comparison of Olive Trees with Different Ploidy Levels

Compared with 5 diploid seedlings derived from open-pollinated seeds of ‘Koroneiki’, ‘Olive-3x’ showed slightly shorter internodes, although the difference was not significant (Figure 5 and Figure 6). Clear morphological differences were observed in leaf shape, as ‘Olive-3x’ produced shorter leaves with a more pointed apex (Figure 5). Measurements further indicated that the leaf area of ‘Olive-3x’ was significantly smaller than that of the 5 diploid seedlings (*p* < 0.01) (Figure 5 and Figure 6).

Anatomical observations of transverse sections of mature leaves were conducted using paraffin sectioning (Figure 7). The triploid exhibited significantly greater total leaf thickness and palisade mesophyll thickness than the diploid seedlings (*p* < 0.01) (Figure 7 and Figure 8). In addition, the upper and lower epidermal cells as well as palisade mesophyll cells of ‘Olive-3x’ were significantly larger than those of the diploids (*p* < 0.01) (Figure 7 and Figure 9).

For the spongy mesophyll, ‘Olive-3x’ exhibited a variable response compared with diploid seedlings (Figure 7, Figure 8 and Figure 9). Spongy mesophyll thickness and cell dimensions were significantly greater than those of several diploid individuals (*p* < 0.05 or *p* < 0.01), while no significant differences were detected relative to some diploids. Overall, variation among diploid seedlings was relatively limited compared with the differences observed between the triploid and diploids. Overall, variation in the leaf traits measured in this study among diploid seedlings was relatively limited compared with the differences observed between the triploid and diploids.

## 3. Discussion

### 3.1. Origins and Characteristics of Triploid Olive

Triploids are typically generated through the fusion of a 2n gamete with a normal gamete, a mechanism well documented in fruit crops such as citrus [32,33] and loquat [34]. In olive, naturally occurring triploids have been reported only in wild populations of Laperrine’s olive [24], and no such cases have been identified in the seed-derived progeny of cultivated olive. According to that study, the triploid frequency in Laperrine’s olive reached 2.8%, which is markedly higher than the 0.036% observed in our study (1 triploid among 2754 seedlings). This difference may reflect the stronger environmental adaptability of triploid Laperrine’s olive, which could confer a survival advantage and result in a higher proportion of triploids in natural populations.

The parentage of the triploid plant identified in this study (‘Olive-3x’) remains uncertain. InDel marker analysis confirmed ‘Koroneiki’ as the maternal parent, as the seeds were harvested from a cultivar-defined plantation. However, since the plantation was managed under open-pollination conditions, ‘Olive-3x’ may have originated from either self-pollination or cross-pollination, which requires further verification. Seedlings of ‘Olive-3x’ exhibited thicker leaves than diploids, likely attributable to larger cell size, a characteristic commonly observed in polyploid species [43,44]. In contrast, the leaf area of ‘Olive-3x’ was significantly smaller than that of diploid seedlings, differing from the larger leaf area reported in tetraploid olives [29]. This discrepancy may reflect developmental differences between juvenile and mature plants, warranting further investigation.

### 3.2. Utilization Potential of Triploid Olive

Triploid plants frequently exhibit advantageous traits across species. Examples include the enhanced vigor reported in triploid poplar [20] and loquat [45], as well as dwarfing characteristics observed in certain triploid genotypes, such as the dwarf mutant ‘A1d’ derived from the triploid apple ‘PYTC’ [45]. Triploids also tend to show improved tolerance to abiotic stresses compared with diploids [46]. These attributes indicate that triploid olives may serve as promising rootstocks for grafting and for use in high-density plantation.

The inherently low fertility of triploids has been widely applied in the development of seedless crops such as loquat, citrus, watermelon, and banana [47,48,49]. However, whether naturally occurring triploid olives are completely sterile, capable of producing fruit, or contain a differentiated endocarp remains unclear. For instance, naturally occurring triploid jujube fruits still retain an endocarp [50], although the proportional changes cannot be precisely assessed due to the absence of diploid controls with identical genotypes. In loquat, triploidy results in seedless fruit with slightly reduced size but an increased proportion of edible tissue, thereby improving fruit usability [51,52]. Whether similar characteristics will appear in triploid olives requires further validation.

Despite their low fertility, triploids can produce parthenocarpic fruit when induced by plant growth regulators, thereby enhancing fruit set. Notably, triploids are not completely sterile; some can produce a limited number of fruits, and their progeny may display diverse ploidy levels, including aneuploids and tetraploids [52,53]. This capability significantly broadens the genetic diversity available for breeding and provides additional opportunities for developing new olive cultivars.

### 3.3. Future Perspectives

The naturally occurring triploid olive identified in this study represents a valuable material for both breeding and basic research. Future studies should focus on evaluating fruit traits, fertility performance, and long-term growth stability of triploid plants under field conditions. In addition, the precise origin of triploid individuals remains to be clarified. The integration of polymorphic molecular markers with higher-resolution genomic approaches, such as Single Nucleotide Polymorphism (SNP) genotyping, haplotype-based parentage analysis, or whole-genome resequencing, would enable more accurate discrimination between self- and cross-pollination events and provide deeper insight into the mechanisms underlying natural triploid formation in cultivated olive. Moreover, combining cytological analyses with molecular and genomic data will contribute to a better understanding of the genetic basis of triploid-associated traits, thereby supporting the effective utilization of naturally occurring polyploid resources in olive improvement and production.

## 4. Conclusions

This study reports the identification of a naturally occurring triploid plant from seed-derived progeny of cultivated olive, providing direct evidence that natural triploids exist within the progenies of cultivated olive varieties. The identification of this triploid demonstrates the feasibility of screening natural polyploid germplasm from seed-derived populations, offering a practical pathway for the discovery of novel polyploid olive resources.

Ploidy determination was achieved through a combination of flow cytometry and chromosome observation, ensuring accurate and reliable identification, while leaf anatomical traits at the seedling stage revealed consistent structural features associated with triploidy. Together, these approaches form a complementary framework that balances accuracy and efficiency, making it suitable for routine ploidy screening in olive.

Furthermore, this study represents the first development and application of polymorphic InDel markers in olive. The established marker set effectively discriminated olive genotypes and clarified genetic relationships, highlighting its utility as a new molecular tool for cultivar identification, parentage analysis, and genetic diversity studies. Collectively, these findings provide methodological support and genetic resources for the exploration of polyploid germplasm and the advancement of olive breeding programs.

## 5. Materials and Methods

### 5.1. Materials

Olive seeds (approximately 3000 in total) were collected from the Fujiang Olive Plantation in the Wudu District of Longnan City, Gansu Province. The batch included open-pollinated seeds of 4 cultivars (‘Leccino’, ‘Picual’, ‘Ezhi 8’, and ‘Pendolino’) (Figure 10), as well as a mixed lot of seeds derived from the plantation’s 12 cultivated varieties but not separated by cultivar. The plantation cultivates the following 12 cultivars: ‘Picual’, ‘Arbosana I’, ‘Arbequina I’, ‘Coratina’, ‘Leccino’, ‘Frantoio’, ‘Pendolino’, ‘Picholine’, ‘Koroneiki’, ‘Golden Leaves’, ‘Ezhi 8# I’, and ‘Yuntai’. One-year-old diploid seedlings derived from open-pollinated ‘Koroneiki’ seeds were purchased from the same plantation in the following year and used as diploid controls.

Young leaf samples from 15 olive cultivars were also collected. Leaves of 12 cultivars were obtained from the Fujiang Olive Plantation in the Wudu District of Longnan City, Gansu Province, whereas leaves of the remaining 3 cultivars (‘Arbosana II’, ‘Arbequina II’, and ‘Ezhi 8# II’) were collected from the Jiangyuan Olive Plantation in the Hechuan District of Chongqing Municipality. Detailed information on the sources of both olive seeds and leaf samples is provided in Table 1.

### 5.2. Methods

#### 5.2.1. Seedling Cultivation

The seeds harvested in the same year underwent cold stratification in sand for three months under conditions of 5–10 °C and 50–70% relative humidity. After a portion of the seeds had cracked, they were sown evenly onto nursery beds with loose, well-drained soil at the experimental field of Southwest University (No. 105 Base), Beibei District, Chongqing, China. During the seedling stage, the nursery beds were covered with a shade net to prevent direct exposure to strong sunlight and to reduce the risk of photodamage to young leaves. A relatively low-density shade net (approximately 30–40% shading) was used, allowing sufficient diffuse light for normal seedling growth while avoiding excessive irradiance. The seedbeds were maintained moist and covered with a thin layer of straw to conserve heat and moisture. Regular soil loosening, weeding, and light applications of nitrogen fertilizer were conducted to promote seedling growth. By September of the following year, a total of 2754 seedlings had emerged.

#### 5.2.2. Polyploid Screening and Identification

(1)Identification by flow cytometry

Newly expanded young leaves were collected from olive seedlings at the 2–4 true-leaf stage for ploidy screening by flow cytometry, following the method described by Liang et al. [54]. To improve screening efficiency, three seedlings were pooled as one sample, and each pooled sample was analyzed in three biological replicates. For each replicate, approximately 1.5 g of young leaf tissue was excised and placed in a Petri dish. The leaf tissue was immersed in 1 mL of precooled nuclear extraction buffer, prepared according to the patented method [55], and rapidly chopped with a sharp razor blade held vertically. An additional 1 mL of extraction buffer was then used to rinse the Petri dish. The resulting suspension was filtered through a 30 μm filter tip into a 2 mL centrifuge tube, followed by the addition of 50 μL DAPI solution (5 μg/mL). After staining in the dark for 3–4 min, nuclear DNA content was measured using a CyFlow^®^ Ploidy Analyser (Sysmex Partec GmbH, Goerlitz, Germany). Genome ploidy levels of olive seedlings were determined and recorded based on fluorescence intensity.

(2)Chromosome counting method

Chromosome slides were prepared using root tips as the material, following the method reported by Liang et al. [54]. Chromosomes were photographed using an Olympus BX53F microscope (Olympus Corporation, Tokyo, Japan).

#### 5.2.3. Identification of Polyploidy Using InDel Molecular Markers

(1)Development of InDel molecular markers using genome resequencing

5 olive cultivars, including ‘Picual’ from the Western Mediterranean genetic pool, ‘Arbequina I’, ‘Koroneiki’ from the Central Mediterranean genetic pool, the ornamental variety ‘Golden Leaves’, and the seedling variety ‘Ezhi 8# I’ selected in Hubei, China, were used for resequencing samples [56,57]. The wild olive genome (*O. europaea* var. *sylvestris*) reported by Unver et al. was used as the reference genome [58]. Whole-genome resequencing was performed using the Illumina NovaSeq sequencing platform, with an expected depth of 20×. InDel molecular markers were developed based on the resequencing results, following the methodology reported by Dai et al. [59].

(2)Validation of molecular markers and screening of polymorphic markers

A total of 460 polymorphic InDel loci, uniformly distributed across the genome with insertion/deletion lengths ranging from 15 to 20 bp, were selected. Primers flanking these loci were designed using Primer 5, with primer lengths set at 18–22 bp. All primers were synthesized by BGI (Shenzhen, China).

The effectiveness and polymorphism of the 460 markers were evaluated using 15 olive cultivars. Twelve cultivars (‘Picual’, ‘Arbosana I’, ‘Arbequina I’, ‘Coratina’, ‘Leccino’, ‘Frantoio’, ‘Pendolino’, ‘Picholine’, ‘Koroneiki’, ‘Golden Leaves’, ‘Ezhi 8# I’, and ‘Yuntai’) were collected from the Fujiang Olive Plantation in the Wudu District of Longnan City, Gansu Province. The remaining three cultivars (‘Arbosana II’, ‘Arbequina II’, and ‘Ezhi 8# II’) were collected from the Jiangyuan Olive Plantation in the Hechuan District of Chongqing Municipality. PCR amplification and electrophoresis were performed following the protocol described by Yan [60].

(3)Analysis of triploids

The amplification results obtained from polymorphic primers were used to analyze the triploid plants together with the 15 reference cultivars. The parental genotypes of the triploids were identified based on the presence of specific diagnostic bands.

#### 5.2.4. Morphological Observation at the Seedling Stage

The main seedling-stage morphological traits, leaf morphology, and leaf anatomical structures of triploids were compared with those of diploid controls. An electronic digital caliper with a precision of 0.01 mm was used to measure internode length, leaf length, and leaf width, and leaf shape characteristics were compared accordingly. Paraffin sectioning was performed to prepare transverse sections of mature leaves following the method of Wen et al. [61]. Leaf thickness, thickness of palisade and spongy mesophyll, and cell size of the upper epidermis, lower epidermis, and palisade tissue were measured using the measurement tools of an OLYMPUS BX53F microscope.

## Figures and Tables

**Figure 1 plants-15-00127-f001:**
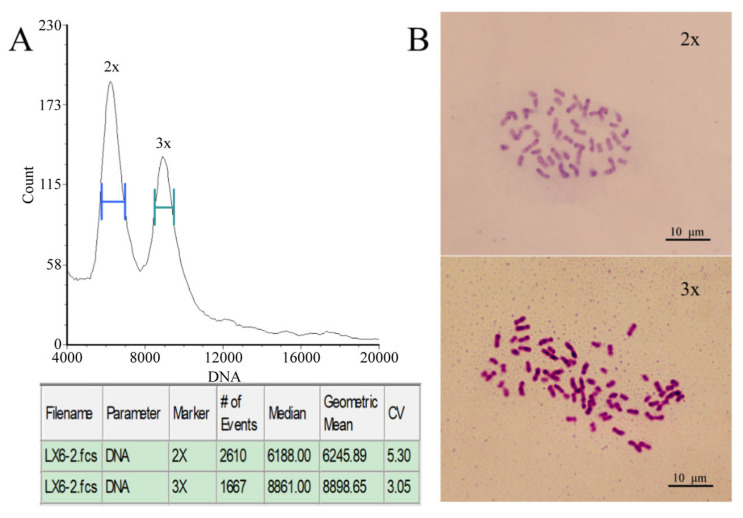
Identification of triploid olive. (**A**) Ploidy determination by flow cytometry. (**B**) Chromosome counting. ‘Koroneiki’ represents the diploid control (2x), while ‘Olive-3x’ represents the triploid (3x).

**Figure 2 plants-15-00127-f002:**
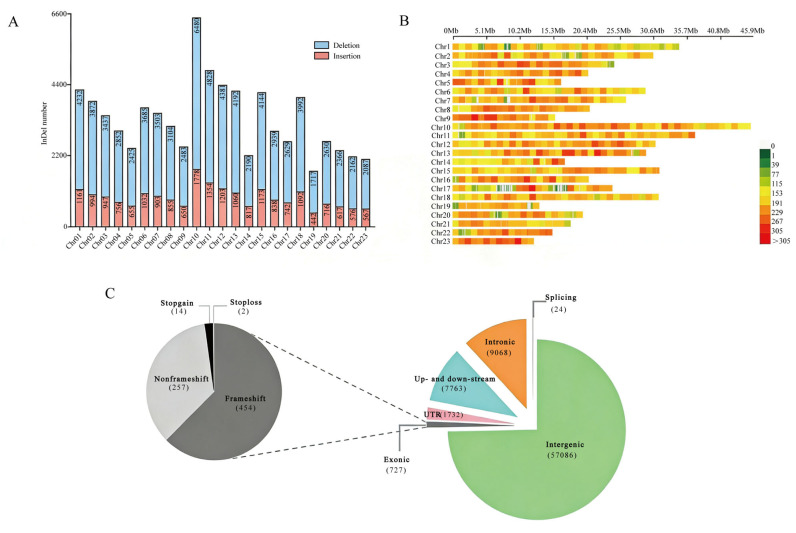
Distribution of InDels (≥15 bp) in the chromosomes of 5 olive cultivars. (**A**) Number of InDels per chromosome. (**B**) Density distribution of InDels across chromosomes. (**C**) Classification of InDels.

**Figure 3 plants-15-00127-f003:**
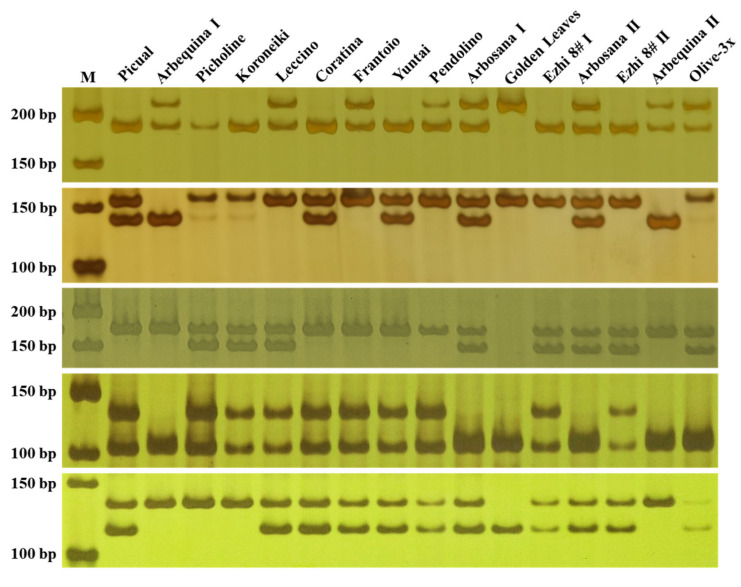
Electrophoresis profiles of PCR amplification using selected primer pairs in 16 olive genotypes. M represents the 500 bp DNA marker. Primer IDs from top to bottom are OE03-11, OE07-7, OE01-12, OE03-18, and OE04-19.

**Figure 4 plants-15-00127-f004:**
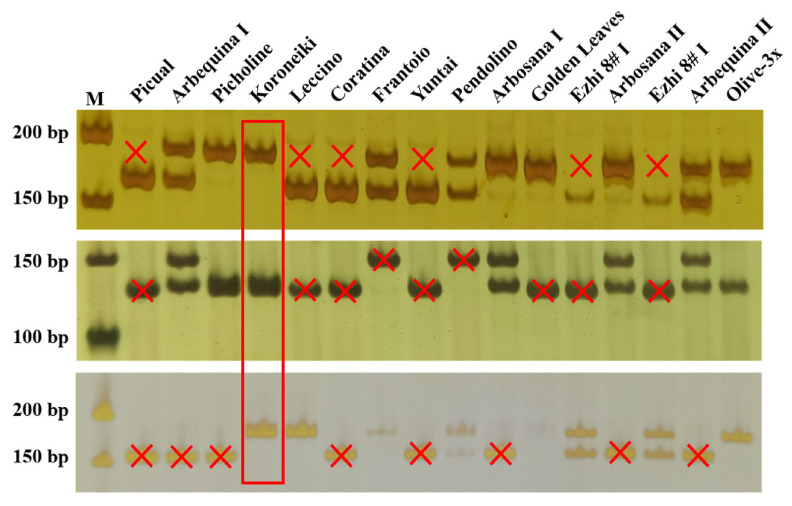
Screening of the potential parent of the triploid using three specific InDel markers. From top to bottom: OE03-5, OE05-12, and OE03-17. M represents the 500 bp DNA ladder. Red “×” indicates the non-parent cultivars excluded due to a single band difference from ‘Olive-3x’, while the red box highlights ‘Koroneiki’ showing identical banding patterns to ‘Olive-3x’.

**Figure 5 plants-15-00127-f005:**
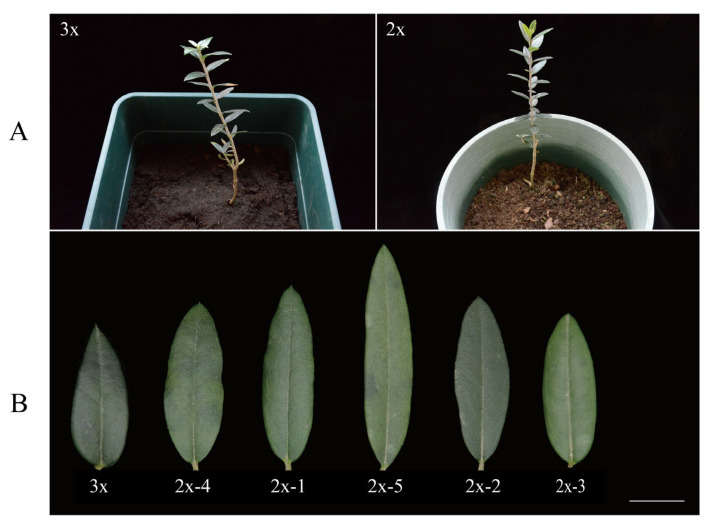
Plants and leaves of ‘Olive-3x’ and 5 diploid seedlings derived from open-pollinated ‘Koroneiki’. (**A**) Whole plants; (**B**) Leaves (scale bar = 1 cm).

**Figure 6 plants-15-00127-f006:**
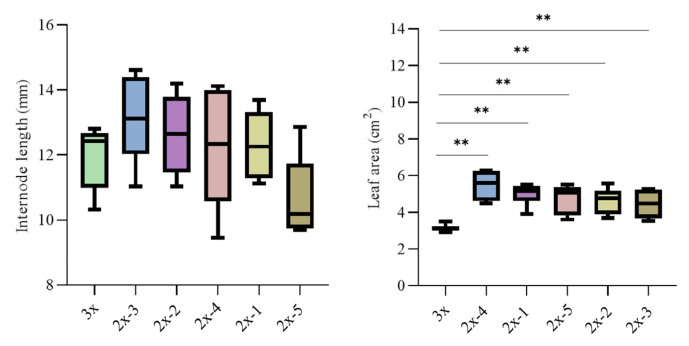
Comparison of internode length and leaf area between ‘Olive-3x’ and 5 diploid seedlings derived from open-pollinated ‘Koroneiki’. ** indicates a highly significant difference (*p* < 0.01).

**Figure 7 plants-15-00127-f007:**
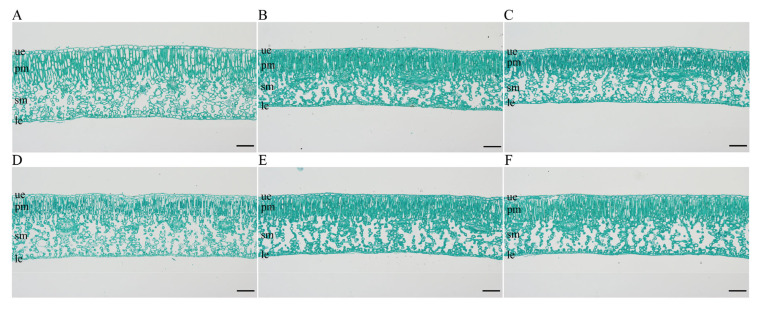
Microscopic anatomical comparison of mature leaves between ‘Olive-3x’ and 5 diploid seedlings derived from open-pollinated ‘Koroneiki’ (scale bar = 100 μm). (**A**) ‘Olive-3x’; (**B**–**F**) 5 diploid seedlings designated as 2x-1, 2x-2, 2x-3, 2x-4, and 2x-5. Leaf tissue types are indicated as follows: ue, upper epidermis; pm, palisade mesophyll; sm, spongy mesophyll; le, lower epidermis.

**Figure 8 plants-15-00127-f008:**
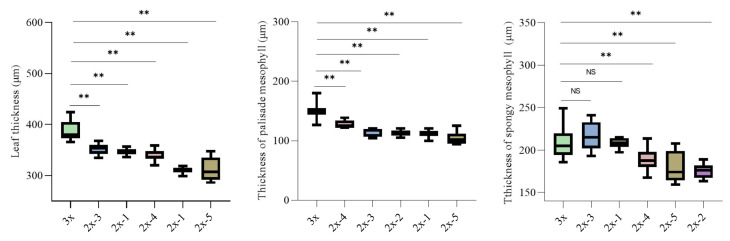
Comparison of leaf thickness, thickness of palisade mesophyll, and thickness of spongy mesophyll between ‘Olive-3x’ and 5 diploid seedlings derived from open-pollinated ‘Koroneiki’. NS indicates no significant difference; ** indicates a highly significant difference (*p* < 0.01).

**Figure 9 plants-15-00127-f009:**
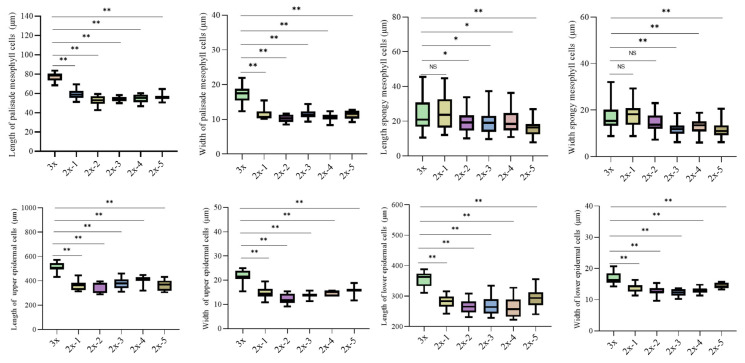
Comparison of cell length and width in the palisade mesophyll, spongy mesophyll, and the upper and lower epidermis between ‘Olive-3x’ and 5 diploid seedlings derived from open-pollinated ‘Koroneiki’. NS indicates no significant difference; * indicates a significant difference (*p* < 0.05); ** indicates a highly significant difference (*p* < 0.01).

**Figure 10 plants-15-00127-f010:**
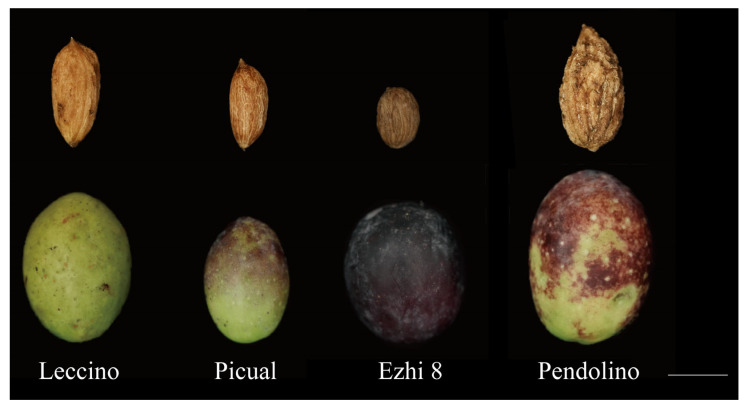
Fruits and seeds of 4 olive cultivars (scale bar = 1 cm).

**Table 1 plants-15-00127-t001:** Source information for olive samples.

Sample	Number	Cultivar	Primary Fruit Use	Source
Seeds	1	Leccino	Oil olive	Fujiang Olive Plantation in the Wudu District of Longnan City, Gansu Province
	2	Picual	Oil olive	
	3	Ezhi 8	Oil olive	
	4	Pendolino	Oil olive	
	5	Mixed-cultivar seeds	-	
Diploid seedlings (OP ‘Koroneiki’)	1	Koroneiki	Oil olive	
Young leaves	1	Picual	Oil olive	
	2	Arbosana I	Oil olive	
	3	Arbequina I	Oil olive	
	4	Coratina	Oil olive	
	5	Leccino	Oil olive	
	6	Frantoio	Oil olive	
	7	Pendolino	Oil olive	
	8	Picholine	Dual-purpose	
	9	Koroneiki	Oil olive	
	10	Golden Leaves	Dual-purpose	
	11	Ezhi 8# I	Oil olive	
	12	Yuntai	Oil olive	
	13	Arbosana II	Oil olive	Jiangyuan Olive Plantation in the Hechuan District of Chongqing Municipality
	14	Arbequina II	Oil olive	
	15	Ezhi 8# II	Oil olive	

The “mixed-cultivar seeds” originated from 12 olive cultivars: ‘Picual’, ‘Arbosana I’, ‘Arbequina I’, ‘Coratina’, ‘Leccino’, ‘Frantoio’, ‘Pendolino’, ‘Picholine’, ‘Koroneiki’, ‘Golden Leaves’, ‘Ezhi 8# I’, and ‘Yuntai’; “OP” means open-pollinated; “Dual-purpose” refers to olive cultivars whose fruits are suitable for both oil production and table use.

## Data Availability

All data generated or analyzed during this study are included in this published article (and its Appendix A).

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
