# Peer review of "Identification of Triploid Plants in Seed-Derived Progeny of Cultivated Olive"

_plants, 2026, doi:10.3390/plants15010127_

Round 1

Reviewer 1 Report

Comments and Suggestions for Authors

The article "Identification of Triploid Plants in Seed-Derived Progeny of Cultivated Olive" by Chenggong Lei, Guangmin Wu, Yingjia Liu, Chengdu Yang, Qianli Dai, Yingchun Zhu, Fa Xiao, Hengxing Zhu, Qigao Guo, and Jiangbo Dang presented the results of molecular cytological identification of the triploid olive genotype.
The manuscript is formatted correctly and contains all necessary parts. The illustrative material is sufficient to prove and identify the mutant form.
Minor comments:
Improve the quality of Fig. 2 – the inscription is illegible.
Please add photos of the seeds and fruits of the olives studied.
Also, please label the leaf tissue types in Figure 6 and discuss the changes in their size in more detail. Judging by the provided photographs, one can see significant differences in the epidermis in both the upper and lower parts of the leaf. Is this true? Can triploid plants be distinguished by epidermal cells or stomata?
Please indicate the illumination conditions and regime in the materials and methods.
Specifically and accurately describe the sample preparation for cytophotometry, taking into account the number of replicates and leaf age.
Please more clearly articulate in your conclusions the advantages of molecular markers for cultivar identification and the methodology of indirect and direct ploidy identification in routine methods.

Reviewer 2 Report

Comments and Suggestions for Authors

The manuscript reported a comprehensive characterization of triploid plants in cultivated olive. The flow cytometry, chromosome counting, molecular marker and phenotypic observation were conducted for the identification of triploid olive plants. The feasibility and reliability of the materials and methods by cytological and molecular methods were clearly presented, and the figures are clear and in good quality. The results are attractive for olive germplasm evaluation, conservation and breeding. It is suitable for the publication after revision considering following suggestions.

  1. Lane 41, europaea should be Olea europaea, since it is appeared the first time of main text.
  2. Lane 49-52, the text of description may have references.
  3. Lane 66. Olea europaea should be O. europaea.
  4. Figure 1A, the 2X and 3X should be corrected as 2x and 3x.
  5. The PCR markers or primers’ sequences should be listed in a supplementary table for the readers to follow.
  6. In Figure 4, only three molecular markers of PCR were used to identify the potential parent, since one of the parents may be heterozygous for providing the 1x gamete. It is suggested to put a brief discussion to have more genetic and genomic methods to precisely determine the origins of the 3x plants.
